# Surrenderers’ Relationships with Cats Admitted to Four Australian Animal Shelters

**DOI:** 10.3390/ani8020023

**Published:** 2018-02-07

**Authors:** Sarah Zito, Mandy Paterson, John Morton, Di Vankan, Pauleen Bennett, Jacquie Rand, Clive J. C. Phillips

**Affiliations:** 1Centre for Animal Welfare and Ethics, University of Queensland, White House Building (8134), Gatton Campus, Gatton, QLD 4343, Australia; s.zito@uq.edu.au; 2Royal Society for the Prevention of Cruelty to Animals Queensland, Brisbane, QLD 4076, Australia; mpaterson@rspcaqld.org.au (M.P.); dstephens@rspcaqld.org.au (D.V.); 3School of Veterinary Science, University of Queensland, Gatton, QLD 4343, Australia; john.morton@optusnet.com.au (J.M.); j.rand@uq.edu.au (J.R.); 4School of Psychology and Public Health, La Trobe University, Bendigo, VIC 3552, Australia; pauleen.bennett@latrobe.edu.au; 5Australian Pet Welfare Foundation, Kenmore, QLD 4069, Australia; jacquie@petwelfare.org.au

**Keywords:** shelter medicine, animal welfare, cat surrender, cat relinquishment, cat semi-ownership, unwanted cat, animal shelter

## Abstract

**Simple Summary:**

The surrender of cats to animal shelters results in financial, social and moral burdens for the community. Human caretaking of cats was explored in a sample of people surrendering cats to shelters in Australia. At the shelters surrenderers classified themselves as owners or non-owners and a questionnaire identified that this was related to their method of acquisition of the cat, their association time with the cat, the closeness of their relationship with the cat and their degree of responsibility for the cat’s care. A model of ownership perception was developed to provide a better understanding of factors influencing ownership perception. Understanding ownership perceptions in cats surrendered to shelters is important as these can inform the development of more targeted and effective intervention strategies to reduce numbers of unwanted cats.

**Abstract:**

The surrender of cats to animal shelters results in financial, social and moral burdens for the community. Correlations of caretaking and interactions with surrendered cats were calculated, to understand more about humans’ relationships with surrendered cats and the contribution of semi-owned cats to shelter intakes. A questionnaire was used to collect detailed information about 100 surrenderers’ relationships with cats they surrendered to four animal shelters in Australia, with each surrenderer classifying themselves as being either the owner or a non-owner of the surrendered cat (ownership perception). Method of acquisition of the cat, association time, closeness of the relationship with the cat and degree of responsibility for the cat’s care were all associated with ownership perception. Many non-owners (59%) fed and interacted with the cat they surrendered but rarely displayed other caretaking behaviours. However, most surrenderers of owned and unowned cats were attached to and felt responsible for the cat. Based on these results and other evidence, a causal model of ownership perception was proposed to provide a better understanding of factors influencing ownership perception. This model consisted of a set of variables proposed as directly or indirectly influencing ownership perception, with connecting arrows to indicate proposed causal relationships. Understanding ownership perception and the contribution of semi-owned cats to shelter intake is important as these can inform the development of more targeted and effective intervention strategies to reduce numbers of unwanted cats.

## 1. Introduction

Unwanted cats surrendered to animal shelters (including welfare organisations and municipal pounds) are a significant burden on the community and create ethical concerns associated with the euthanasia of healthy animals, moral stress for the people involved, financial costs to organisations that manage unwanted cats and welfare concerns for the cats [1,2,3,4]. Many unwanted cats also live in the community, resulting in other problems such as predation on wildlife, cat to cat and inter-species disease transmission (including zoonotic disease), nuisance behaviours, cat welfare issues and perpetuation of the breeding of additional unwanted cats [2,5]. Many of these cats are supported to varying extents by humans who intentionally provide some level of care (food, medical treatment or shelter) but do not perceive themselves as owners of the cats [4,6]. The term “semi-owned” has been proposed for these cats to distinguish them from cats that are not receiving direct support from humans and from owned cats (those that have a human caretaker who provides for the cat’s needs and may or may not accept ownership and full responsibility for the cat) [7]. Semi-owned cats, by virtue of their association with carers, usually develop some degree of sociability but, in Australia at least, they are usually left sexually entire [6].

The Australian Royal Society for the Prevention of Cruelty to Animals (RSPCA) is Australia’s largest animal sheltering organisation. On admission to RSPCA animal shelters in Australia, cats are classified as either “owned” or “stray,” the latter indicating that the person surrendering the cat does not identify themselves as the cat’s owner or their agent. A high proportion of shelter cat admissions from the general public are classified as “strays” with 54–82% classified as “strays” in Australia [4,8] and 31% in the United Kingdom [9]. In the United States, currently 27% of cats obtained as pets are strays, down from 35% in 2012 [10]. RSPCA shelter staff do not differentiate between semi-owned cats and cats that are not receiving direct support from humans. It is likely that semi-owned cats constitute a considerable proportion of the “stray” cats surrendered to shelters in Australia, based on the sociability and relatively good physical condition of surrendered cats [4,8] and the frequency of semi-owned cats in the community [6]. However, direct evidence from surrenderers has not been reported.

The distinction between owned cats, semi-owned cats and cats that are not receiving direct support from humans is important because different strategies to reduce the numbers of cats admitted to shelters are required for each population. Legislation requiring sterilisation, identification and cat curfews will only have an impact on the owned cat population and then only with the compliance of owners. Strategies such as removal and trap and euthanasia programs aimed at cats that are not receiving direct support from humans may be ineffective for semi-owned cats as the success of these programs is dependent on community support [11]. Cat caretakers, such as semi-owners, are unlikely to support such programs [6,12]. People who do not perceive themselves as the owners of cats they feed and/or provide basic care for may, however, be responsive to education, social marketing messages, or other programs aimed at reducing this population of cats through non-lethal means.

The psychological construct of ownership is complex; categorisation of cats as owned or “stray” oversimplifies the spectrum of human-cat relationships. People may show similar caretaking behaviours towards and interactions with cats despite some perceiving themselves as the owner and others not. Ownership perception may influence the behaviours people show towards cats. Alternatively, caretaking behaviours may influence the caretaker’s perception of ownership. Surrenderers’ caretaking behaviours towards and interactions with owned and unowned cats they surrender have not been previously investigated. Understanding ownership perception and factors that underpin it (such as caretaking behaviours and interactions) may provide the insight necessary to develop novel approaches to effectively reduce the problem of unwanted cats. For outcomes caused by the collective effects of multiple factors, the underlying causal patterns can be complex and explicit conceptual models of causality can inform research planning and interpretation and assist in planning interventions. Where there are multiple direct causes, each of these direct causes can, in turn, be caused by preceding factors. These preceding factors are of interest as they can indirectly affect the occurrence of the outcome of interest, mediated through their effects on direct causes. Some preceding factors may also directly affect the specified outcome of interest. Interventions that target preceding factors (rather than the outcome directly) may be chosen in an attempt to indirectly affect the specified outcome of interest. Directed acyclic graphs [13,14,15,16] are a method developed relatively recently for depicting proposed direct and indirect determinants of specified outcomes of interest, along with proposed interrelationships amongst those determinants. The outcome of interest and each proposed determinant are displayed in a 2-dimensional diagram, with one-directional arrows depicting proposed causal pathways both directly to the outcome of interest and to other proposed determinants. It is important that conceptual models of causality are coherent with empirical evidence and theoretical logic.

To successfully study ownership perception in people surrendering cats to animal shelters, it is necessary to solicit information from a broad range of surrenderers. It can be difficult to recruit people to be participants in studies investigating animal surrender [17,18,19], cat semi-ownership [20,21] and animal caretaking practices [22,23], because of people’s reluctance to discuss these issues. As a consequence, it is difficult to obtain representative samples [20,21,22]. People are possibly more likely to participate if they have some interest in the outcome, with pet owners or caretakers who are interested, committed, and/or responsible probably being more motivated to participate [22]. Nevertheless, studies using small and potentially non-representative samples can provide useful preliminary information and guide further research in this complex and poorly understood area.

We therefore conducted a pilot study that used a convenience sample to describe surrenderers’ caretaking behaviours towards and interactions with, the cats they surrender and to assess associations between potential determinants of perceived ownership (factors such as surrenderer demographics, association time, acquisition method and attachment) and the surrenderer’s perceived ownership of the cat at the time of surrender. Based on these results, other sources of evidence and theoretical logic, a model of the causal inter-relationships for causes of perception of ownership of cats at the time of surrender was proposed. Potential determinants of feeding unowned cats were also assessed.

## 2. Methods

### 2.1. Study Design

The study design and participant details have been previously described in full [7]. The study was approved by the University of Queensland Ethics Committee (project number 2011001160). In brief, people surrendering cats to four RSPCA shelters in Australia between 1 February and 30 September 2012 were asked by admission staff if they would participate in the study. Data were subsequently collected from 141 consenting participants (‘surrenderers’) using a standardised questionnaire administered by telephone interview (*n* = 128) or through a web site (*n* = 13) between one and eight weeks after the surrender that resulted in enrolment. Both forced choice and open-ended questions were used to obtain data about the surrenderer’s relationship with the surrendered cat and their caretaking behaviours towards and interactions with the cat (Table 1).

Data specific to each cat surrendered at enrolment (for example, whether the cat was fed by its surrenderer) and corresponding surrenderer-level data (for example, the surrenderer’s demographic data) were included in analyses. The questionnaire categories and details of variables derived from the questions are provided in Table 1. All responses were entered directly into a digitised questionnaire [24].

Each cat was classified as either owned or unowned by the surrenderer, based on the surrenderer’s level of agreement on a five point Likert scale with the statement “I consider myself to be the owner of the cat.” As the distribution of these responses was bimodal and highly polarised, with few responses in the middle of the scale, the Likert scale responses were converted to a dichotomous variable for analyses; cats with a surrenderer who strongly or somewhat agreed with the statement were considered owned by the surrenderer and cats with a surrenderer who did not agree (surrenderers who neither agreed nor disagreed, or strongly or somewhat disagreed with the ownership statement) were considered to be not owned by the surrenderer. For cats classified as owned by the surrenderer and cats classified as not owned by the surrenderer, the surrenderer was respectively referred to as the ‘owner’ or the ‘non-owner.’

Acquisition method for each surrendered cat was defined based on the source of the cat; cats were either passively acquired (i.e., the cat was found, originally a “stray”, a gift, brought home by children or left with the surrenderer by another person) or actively acquired (i.e., the cat was acquired intentionally by the surrenderer from an animal shelter, breeder, pet shop or through a private transaction). Association time was defined as the duration of the time period for which the surrenderer had an association with the surrendered cat. This was determined for owners based on their response to the question “How long did you own the cat for?” and, for non-owners, based on their responses to the questions “Where was the cat before you took it to the shelter?” and “How long had the cat been there before you took it to the shelter?” Association times were then altered for non-owners if they indicated they had been feeding the cat for longer than stated and for both owners and non-owners if any other data indicated an alternative association time.

Surrenderers who had been associated with the surrendered cat for less than three days were considered ineligible for the study as this was deemed to be insufficient for a relationship, caretaking behaviours and interactions to establish. Some surrenderers surrendered multiple cats on the day they were asked to participate in the study. Since clustering of responses for cats from the same surrenderer was likely, we ensured cats were statistically independent of each other by randomly selecting one cat for analyses from each of these surrenderers, using computer-generated random numbers.

### 2.2. Statistical Analyses

#### 2.2.1. Pooling of Data

Distributions of key variables (association time, ownership perception and cat acquisition method, caretaking behaviours and interactions) were compared by data collection method (questionnaire administered by telephone interview or through a web site). As there were no major differences between these sub-groups in distributions of the key variables assessed [7], data from the two collection methods were pooled for analyses. Responses for adult cats (*n* = 113) and kittens (*n* = 28) were also similar, based on previous analyses [7], so these groups were pooled and, hereafter, adult cats and kittens are collectively referred to as cats. Participants from all shelters were pooled for analyses. Statistical analyses were performed using Stata12^©^ (version 12, StataCorp, College Station, TX, USA). All *p*-values were two-sided.

#### 2.2.2. Correlations between Surrenderers’ Cat Caretaking Behaviours and Interactions

Pair-wise correlations between the nine caretaking and interaction behaviours were assessed using phi correlations and Fisher’s exact tests, both calculated using Stata’s-tabulate-command.

#### 2.2.3. Determinants of Perceived Ownership at the Time of Surrender

Associations between perceived ownership of the cat at the time the cat was surrendered and each of cat acquisition method, association time and surrenderer’s relationship with, number of types of caretaking behaviours shown towards and number of types of interactions with, surrendered cats were assessed. Where these were binary data, Fisher’s exact tests was used, calculated with Stata’s-tabulate-command. Where these were ordinal data, Mann-Whitney rank-sum tests were used, calculated with Stata’s-ranksum-command.

Associations between potential determinants of ownership and perceived ownership of the cat at the time the cat was surrendered were assessed using only surrenderer surrendering passively acquired cats. Nineteen potential determinants of ownership were assessed: surrenderer’s association time, gender, age, occupation, their postcode’s socioeconomic indices and whether they had previously owned a cat, their attitudes towards cats in general (Table 1) and for the surrendered cat, whether the cat was an adult or kitten, the surrenderer’s degrees of responsibility for and attachment to the cat (Table 1), whether the surrenderer fed the cat, the number of types of other caretaking behaviours shown towards the cat and the number of types of interactions with the cat. Each variable was assessed adjusted only for association time. This variable had been shown previously to be a strong determinant of perceived ownership [7] and so could have caused confounding of relationships between other putative determinants and perceived ownership. Only passively acquired cats were used as perceived ownership varied only within passively acquired cats; all surrenderers with actively acquired cats considered that they owned the cat. We categorised each continuous exposure variable into three categories before performing analyses, choosing category cut points that both ensured at least modest numbers of cats in each category and that were ‘neat’ (e.g., for association time, 1 month and 12 months were used as cut points, while for surrender’s age, 35 and 55 were used). Associations were assessed using logistic regression, with the -logistic- command in Stata. Maximum likelihood logistic regression was used, except when one or more combinations of putative determinant category and perceived ownership status had no cats, in which case exact logistic regression was used, with the -exlogistic- command in Stata. For the exact models, conditional probability tests and median unbiased estimates were used. The mid-*p*-value rule was used as recommended by Agresti [28] except for 95% confidence intervals that extended to positive infinity. For the maximum likelihood models, the overall significance of exposure variables was assessed using likelihood ratio tests. For the exact models, all exposure variables fitted had more than two levels; the overall significance of these variables was assessed using joint-significance hypothesis tests. Due to the limited sample size, more complex multivariable models were not fitted.

#### 2.2.4. Model Development

A model was developed to diagrammatically represent hypothesised causal inter-relationships for causes of perception of ownership of cats at the time they are surrendered. This was designed with structural similarities to directed acyclic graphs [13,14,15] but we depicted feedback loops (e.g., where, for example, variable A affects variable B, which in turn, affects variable A) as two-directional arrows. This was done solely to simplify the appearance of the model. In directed acyclic graphs, feedback loops could instead be incorporated by showing variable A at time point 1 affecting variable B at time point 2, which in turn, affects variable A at time point 3 and so on. In our model (as with directed acyclic graphs), each pathway indicated our hypothesis that one variable causally affected another. For example, if variables A and B were binary data (yes/no), a pathway from variable A to variable B would indicate that, if a cat was exposed to (i.e., ‘yes’ for) variable A, that exposure was hypothesised as causing an increase (or decrease) in the probability that it was yes for variable B. If instead variable A was continuous data, if a cat had a higher value for variable A, that would cause a higher (or lower) probability that it was yes for variable B than if it had a lower value for variable A. And if instead variable B was continuous data, if a cat was exposed to (i.e., ‘yes’ for) variable A, that exposure would increase (or decrease) the cat’s expected value for variable B.

To avoid including variables in the model that were unlikely to be causal, other than association time and surrenderer’s gender, only those variables with overall *p*-values of 0.02 or less from our logistic regression analyses were included. Association time and surrenderer’s gender were included because of strong a priori evidence that they are involved in determining ownership, supported by the estimated odds ratios from our logistic regression analyses. We placed variables with no proposed causes of interest to us to the far-left-hand side, ownership perception (the ultimate dependant variable) to the far right-hand side and other variables to be included between these. Thus, causal sequences would generally flow temporally from left to right. We then decided which variables to place pathways from and to and the directions of effect for each of these pathways. Both direct pathways (i.e., pathways ending at ownership perception) and pathways to other variables were allowed. Pathways (relationships) were hypothesised based on published literature and other evidence. Justifications for each proposed relationship are detailed later in this paper.

#### 2.2.5. Associations with Feeding Unowned Cats

Associations between each potential determinant of feeding and whether or not the surrendered cat was fed were assessed in unowned cats with exact logistic regression, using the same potential determinants and statistical methods as described above for ownership. Shelter was fitted in all models as unowned cats surrendered to shelters 1 and 3 were markedly more likely to have been fed (65% or 20/31 and 100% or 5/5, respectively) than for shelter 3 (29% or 5/17). There was no evidence of strong association between the time period for which the surrenderer had an association with the surrendered cat (association time) and whether the respondent fed the surrendered cat, so this was not fitted as a covariate when analysing other potential determinants of feeding.

## 3. Results

There were 141 surrenderers in total, of which one was excluded because data were not reported for most cat level questions (and the number of cats surrendered was not reported). The remaining 140 surrenderers surrendered 177 cats, of which 5 surrenderers, each with 1 cat, were excluded because association times for the 5 cats (and hence eligibility for the study) could not be determined. A further forty-three cats were ineligible for the study as the surrenderer’s association time with them was less than three days. This left 96 surrenderers who surrendered a total of 129 cats on the day the surrenderer was asked to participate in the study: 76 surrenderers surrendered one cat, 10 surrenderers surrendered two cats, nine surrenderers surrendered three cats and one surrenderer surrendered six cats. One cat from each of the 20 surrenderers who surrendered multiple cats was randomly selected, excluding a further 33 cats. Thus, 96 surrenderers, each with one cat, were used in analyses; 40 of these cats were classified as owned and 56 as non-owned.

Descriptions of cat acquisition method, association time and surrenderer’s relationship and interactions with and caretaking behaviours towards, the surrendered cats are detailed in Table 2. Distributions of most variables varied markedly between owners and non-owners. For owners, two-thirds of the cats had been actively acquired but none of the cats surrendered by non-owners had been actively acquired. Almost half of the owned cats (48% or 19/40) but only 11% (5/56) of the unowned cats had been associated with the surrenderer for 12 months or more. The majority of surrenderers (98%; 39/40 of owners and 70%; 39/56 of non-owners) considered that they were responsible for the cat’s care. Most owners (75%; 30/40) but only a third of non-owners (31%; 17/54) felt that they had a close relationship with the surrendered cat (Table 2).

For each of the four types of interactions assessed (Table 2), the majority of owners (76–100%) had that interaction with the cat. Some non-owners also had these interactions with the cat but this was less common among non-owners (7–35% depending on specific interaction) (Table 2). Of the 37 owners who supplied data for all four types of interactions, 78% (29) had all four types of interactions with the surrendered cat compared with only 8% (4/52) for unowned cats. However, 40% (21/52) of unowned cats had at least one of these four types of interaction.

Most owners but few non-owners contained the surrendered cat to their property at night (76%; 28/37 and 14%; 7/51, respectively) (Table 2). Owners more commonly bought toys for the cat (84%; 31/37) than having the cat sterilised (27%; 8/30), vaccinated (74%; 28/38), or microchipped (59%; 22/37) but caretaking behaviours were uncommon among non-owners except for feeding; 57% (31/54) of non-owners fed the cat they surrendered. The majority of these (90%; 27/30) had fed the cat 6 or 7 days every week, had done so for one month or more (60%; 18/30) and had bought food specifically for the cat (57%; 17/30; Table 2). Of the 35 owners who supplied data for the four types of caretaking behaviours other than feeding and sterilisation; (Table 2), only 54% (19) had shown all four behaviours towards the surrendered cat, compared with 0% (0/41) for unowned cats. Of the 34 owners who supplied data for these eight types of interactions and caretaking behaviours, 88% (30) implemented at least four compared to 11% (5/46) for unowned cats. However, 37% (17/46) of unowned cats had at least one of these eight types of interactions and caretaking behaviours. When feeding was included, 60% (27/45) of unowned cats had at least one of these nine types of interactions and caretaking behaviours.

### 3.1. Correlations between Surrenderers’ Cat Caretaking Behaviours and Interactions

Other than ‘Having the cat sterilised,’ the various caretaking behaviours and interaction variables were moderately closely associated with each other (phi coefficients 0.42 to 0.93; Table 3). (Phi coefficients can be interpreted in a similar way to Pearson’s correlation coefficients.) For example, of the 33 cats confined to the surrenderer’s property at night (with or without being confined during the day), the majority (25 or 76%) were also vaccinated and of the 53 cats not confined to the surrenderer’s property, few (only 3 or 6%) were vaccinated (phi coefficient = 0.73).

Having the cat sterilised and feeding were generally less closely associated with other caretaking behaviours and interactions with the cat (phi coefficients 0.15 to 0.36 and 0.15 to 0.62, respectively) due to cats receiving other caretaking behaviours or interactions but not being desexed and due to unowned cats being fed but not receiving other caretaking behaviours or interactions. (All owned cats were assumed to have been fed.) For example, of cats that were fed, 81% (38/47) were not sterilised, 46% (30/65) were not confined to the surrenderer’s property at night, 55% (36/65) were not vaccinated and 64% (41/64) were not microchipped.

### 3.2. Determinants of Ownership Perception at the Time of Surrender

Of the 96 surrenderers, each with one cat, used in analyses, for 71, the cat was passively acquired. Associations between potential determinants and ownership perception of the cat at the time of surrender among these 71 passively acquired cats are described in Table 4. Only passively acquired cats were used as perceived ownership varied only within passively acquired cats. Of the 71 surrenderers and cats where the surrendered cat had been acquired passively, for 21% (15/71) of these, the surrenderer perceived themselves to be the owner of the cat, whereas of the 25 surrenderers that had surrendered a cat they actively acquired, all perceived themselves to be the owner of the cat.

Of cats surrendered to shelters 1, 2, 3 and 4, 16% (6/38), 100% (1/1), 6% (1/18) and 54% (7/13), respectively, were perceived to be owned. After adjusting for association time, shelter was associated with ownership (overall *p* = 0.01), largely due to the high prevalence of perceived ownership amongst cats surrendered to shelter 4. When assessing other potential determinants, the data were too sparse to simultaneously adjust for both association time and shelter. As 65% (46/71) of cats had association times where the prevalence of perceived ownership was higher (association times of 1 month or longer) compared to only 18% (13/71) of cats surrendered to shelter 4, association time was potentially the more important confounder, so when assessing other potential determinants, we adjusted for association time but not shelter.

Other than association time and surrenderer’s gender, only those determinants with overall *p*-values of 0.02 or less when adjusted for association time are included in Table 4. The other eleven potential determinants that were assessed had overall *p*-values when adjusted for association time of 0.34 to 0.96. Surrenderer attributes most strongly associated with ownership perception of the surrendered cat (all *p*-values < 0.01, estimated odds ratios >4 or <0.25) after adjustment for association time were agreement of surrenderer with the statement “I considered myself to be responsible for the cat’s care,” agreement with the statement “I had a close relationship with the cat,” feeding the cat, displaying one or more types of caretaking behaviours other than feeding or sterilisation towards the cat and having one or more types of interactions with the cat (Table 4). Surrenderers aged over 55 were less likely to identify themselves as the owner of the surrendered cat compared to younger surrenderers.

Association time and surrenderer’s gender are included in Table 4 because they were selected along with other variables from Table 4 for inclusion in a hypothesised model of the causal inter-relationships for perception of ownership of cats at the time the cat was surrendered (Figure 1). The bases for each of the pathways included in the conceptual model are presented in Table 5. Association time and surrenderer’s gender were included because of strong a priori evidence that they are involved in determining ownership (Table 5), supported by the estimated odds ratios as reported in Table 4.

### 3.3. Associations with Feeding of Unowned Cats

Of the 56 unowned cats (all of which were passively acquired), feeding status (whether or not the surrenderer fed the cat) and shelter were recorded for 53. Associations between potential determinants and feeding the surrendered cat amongst these 53 unowned cats, where the overall *p*-value adjusted for shelter was ≤0.13, are described in Table 6. The other eleven potential determinants of feeding that were assessed had overall *p*-values adjusted for shelter of 0.26 to 0.96.

Variables most strongly associated with feeding of unowned cats (all *p* < 0.05, estimated odds ratio >4) after adjustment for shelter (Table 6) were agreement with the statement “I considered myself to be responsible for the cat’s care” and having more than two types of interaction with the cat. There was also some evidence that females are more likely than males to feed unowned cats, that kittens are more likely to be fed than adult cats and that surrenderers who display one or more types of caretaking behaviours other than feeding or sterilisation towards the cat are also more likely to feed the cat.

## 4. Discussion

Most owners in the study displayed a variety of types of interactions with the cats they surrendered but purposeful caretaking behaviours, such as containment at night, vaccination and microchipping were less common (59–76% of owners) and only 27% of surrenderers with association times ≥1 month had the cat sterilised. These low prevalences of caretaking behaviours may be because owners who surrender their cats are less likely to show purposeful caretaking behaviours than owners who retain their cats. This is supported by previous research showing that the prevalence of sterilisation is higher among owned cats than owner-surrendered cats [4,6,8,38,42].

Although the frequency of caretaking behaviours towards and interactions with, unowned cats was lower than for owned cats, many non-owners displayed caretaking behaviours towards and interactions with unowned cats. The majority (60%) of non-owners displayed at least one caretaking behaviour (disregarding sterilisation) towards, and/or interaction with, the surrendered cat. Among non-owners, feeding was the most common caretaking behaviour; other caretaking behaviours were not observed without feeding and only four non-owners interacted with the surrendered cats but did not feed them. Of the 57% of non-owners who fed the cat they surrendered, the majority had done so almost every day for at least one month. Consequently, over half of non-owners may be classified as semi-owners, based on the definition of semi-owners as people who intentionally provide food, medical treatment or shelter to cats that they do not consider that they own [6]. Cat semi-ownership results in potential welfare issues for cats, such as supplying food that is variable in quantity and quality, attracting cats to high traffic areas and encouraging the production of unwanted kittens, which together with the original cat may be surrendered to a shelter [6,8]. This phenomenon must be addressed in any plan to try and improve cat welfare and decrease the number of unwanted cats. Therefore, it would be valuable for shelters to attempt to classify the ownership status of cats on admission and to quantify the proportion of cats admitted that are semi-owned. Such knowledge would enable welfare groups to make better informed decisions about the use of their available resources for effectively targeting unwanted cat numbers and shelter intakes in their area. Questions about the caretaking behaviours shown towards surrendered unowned cats could be added to the mandatory data collected at point of entry to the shelter, to provide data which can be used to determine if the cat is semi-owned. However, it must be acknowledged that lengthening the intake form may reduce the accuracy of the data, which has been questioned for companion animals being surrendered to shelters on the grounds of the owner’s desire to provide a socially acceptable reason for surrender, or to improve the chances of admittance to the shelter and subsequent adoption [43]. Fees may be charged for surrender of owned cats but not strays, which may also affect data accuracy. If cat semi-owners are making a considerable contribution to shelter intake then specific programs targeting cat semi-owners, for example education campaigns and the provision of low cost or free sterilisation services, could be introduced into the shelter’s community to reduce the number of unwanted cats.

Most owners and many non-owners reported being attached to the cats they surrendered, which suggests that pressures such as financial, health or family problems (as distinct from lack of attachment) may have resulted in the surrender. Many owners surrendering their cats have been reported to have high levels of attachment to the cats and struggle with the decision to surrender [21,44]. Cat semi-owners and cat colony caretakers have also been reported to be attached to the cats they care for, indicating that they feel protective of the cats and that the cats are “like pets” [12,45]. Consequently, it might be possible to reduce numbers of cats surrendered if the problems leading to the surrender can be addressed. Attachment and affection towards cats (both owned and semi-owned) also means that their caretakers are likely to be opposed to the idea of lethal means of control [12,45,46,47] and, therefore, may obstruct lethal cat management strategies that target “stray” cats.

Non-owners who were more likely to feed the surrendered cat were people who felt they were responsible for the cat’s care, people who displayed one or more caretaking behaviours towards the cat (other than feeding) and people who had interacted with the cat. Despite the lack of perception of ownership among people who fed unowned cats, the attributes associated with the feeding of unowned cats were all attributes which were also associated with ownership perception. However, the fact that some non-owners fed the surrendered cat indicates that there are determinants of feeding in addition to those factors that determine ownership perception. Greater understanding of the factors contributing to people feeding cats they do not perceive ownership for could help to inform education campaigns to try and encourage other caretaking behaviours, such as sterilisation, which may help to reduce the number of unwanted kittens born to semi-owned cats. 

All actively acquired surrendered cats were considered owned compared with 21% of passively acquired cats and, therefore, acquisition method was a key determinant of perceived ownership. Length of association also had a strong positive association with ownership. Ownership perception is a complex concept and one that will be influenced by interactions between many different factors. However, contrary to expectations from the extant literature on cat ownership (Table 5), women were not more likely to claim ownership of passively acquired cats than men (Table 4). This may be due to their sense of deference to the attitudes of men on this issue, a factor previously acknowledged in women’s’ attitudes to animal welfare [48]. In this respect, attitudes of men may be more negative towards their female partner owning passively-acquired cats than actively-acquired cats.

A model for perception of ownership was proposed that represents the complex inter-relationships between the putative determinants of ownership perception in surrenderers based on analyses of the study data, published literature and other evidence. This could provide a valuable foundation for future investigations. Each pathway in the proposed model was informed by logic and evidence from the literature but the model requires further testing for validity. A greater understanding of causes of perception of ownership of cats could not only inform potential intervention points for animal welfare groups trying to promote cat ownership and prevent cat semi-ownership and surrender but would also lead to a clearer understanding of people’s relationships with cats. For example, assuming our hypothesised causal model is correct, to increase the probability that people with passively acquired cats will consider themselves to be the owner of the cat, strategies that encourage these people to interact in more ways with the cat and to take responsibility for the cat should be considered. If more types of interactions result and/or if more responsibility is taken, such strategies would be expected to also result in display of more caretaking behaviours toward the cat.

### Limitations of the Study

Although several attributes associated with perceived ownership were identified, it is likely that there are additional determinants. These could be that the person has not owned an animal before (and so may not have a good understanding of the concept of ownership), is unwilling to take on the responsibility of cat ownership, is ethically opposed to ownership, does not feel they are capable of owning an animal, does not feel they have the resources required, think that the cat may belong to someone else or simply because they did not pay for or actively acquire the cat. This identifies an important area for future research to further elucidate the psychology unpinning ownership perception, especially in people who feed cats for whom they do not perceive ownership (semi-owners), since this group of people contribute to the number of cats presented to shelters as “strays” as well as to cat welfare issues and cat overpopulation [6,8]. 

A cross sectional design was used for this study and hence ownership perception and attachment were measured only at the point in time when the cat was surrendered. Caretaking behaviours and interactions up to that time were assessed but the dates of occurrences of these were not assessed. Thus, the temporal order of events leading to ownership perception and the direction of causal relationships can only be postulated. A longitudinal panel study where ownership perception, attachment and caretaking behaviours and interactions are repeatedly assessed over time in the same people would help to determine time sequences for the occurrence of these events and how they interact, better informing causal understanding. It is likely that there are iterative effects within many of the relationships, in which attachment, caretaking behaviours and interactions and ownership perception are dynamic and influenced over time by each other and by positive or negative feedback from the cat’s behaviour. For example, if a person feeds a new, apparently unowned cat, the cat may become more friendly and approachable. Because of this, the person may then start to interact with the cat, which will positively reinforce the developing relationship, leading to more complex interaction and a greater chance of the person perceiving ownership. In contrast, if a cat is fearful or aggressive, interaction and some caretaking behaviours (such as vaccination and others that require handling) are probably less likely, with consequent effects on ownership perception. The proposed model provides a foundation for building better understanding of causes of ownership perception and could act as the basis for future research. The model could be developed further by addition of potentially important factors that might influence ownership perception such as education about cat ownership/care. Elements from the Theory of Planned Behaviour [49] might also be added, including attitudes toward various cat behaviours, subjective norms and perceived behavioural control. Longitudinal panel studies, with large numbers of respondents assessed repeatedly over time, would be needed to test and validate the model.

As with all studies in this field, difficulties in recruiting participants for this study resulted in both a relatively small number of participants and uncertainty about the representativeness of the study population in relation to all cat surrenderers. It is difficult to recruit people to discuss the surrender of a cat [17,18], probably because many people are upset at the loss, or feel guilty and so do not want to participate. Those who do consent are likely to represent a biased sample; it seems likely that those who care enough about the cat and/or cats in general to overcome their guilt or distress for the greater good are more likely to participate. All research that relies on voluntary participation to answer questionnaires faces the same problems. Research findings from such studies nevertheless can provide valuable empirical data that can guide more detailed investigations. This study, although pilot in nature, has produced empirical evidence that substantiates previous suggestions that a substantial proportion of cats admitted to shelters as “strays” are in fact semi-owned [4,8]. Larger studies using more representative samples of surrenderers are needed to further investigate humans’ relationships with surrendered cats and the contribution of semi-owned cats to shelter intakes. Larger sample sizes would allow use of more complex multivariable statistical models and more representative samples would reduce risk of selection bias and also increase external validity. However, such samples may be impossible to obtain using standard research methodologies that, for ethical reasons, demand voluntary participation. The routine collection of more specific information for all surrendered cats could be achieved as part of the obligatory shelter admission procedures and may enable researchers to circumvent some of the issues inherent in data collection in this research area. The results presented in this study could be used to guide the framing of additional routine admission questions. For example, recording additional information from surrenderers—about association time, cat feeding and frequency and other caretaking behaviours shown to the cat—would provide enough information for conclusive answers to build on the findings from this study. Such information would also provide an estimate of the contribution of semi-owners to shelter intake and this could be valuable at a local level, enabling each shelter to implement strategies that are more likely to be effective in their specific geographic area.

Our model of ownership was based, in part, on our results from analyses of data from cats that were surrendered and it is quite possible that a different model explains how perceived ownership is caused amongst cats that are not surrendered, particularly amongst passively acquired cats. The psychological processes involved in a person choosing to surrender a passively acquired cat are likely to be complex and also interrelated with causes of perceived ownership. Accordingly, further research should be conducted using passively acquired cats that are not surrendered before our model is applied to such cats.

## 5. Conclusions

Among people surrendering cats to RSPCA shelters in Australia, active acquisition of cats strongly predicts perceived ownership status and association time is also strongly and positively associated with ownership perception. Many of the cats being surrendered to RSPCA shelters as unowned are “semi-owned.” Most owners and many non-owners feel responsible for and are attached to the cat they surrender.

Identified determinants of ownership perception—acquisition method, association time, a close relationship with the cat, responsibility for the cat’s care, age of surrenderer and increasing numbers of caretaking and interaction behaviours shown to the surrendered cat—were used to develop a model for perception of ownership. The model highlights the complexities of ownership perception and provides a foundation for building a better understanding of humans’ relationships with surrendered cats. This, in turn, may inform more targeted and effective intervention strategies to reduce cat intake into shelters and promote cat welfare.

## Figures and Tables

**Figure 1 animals-08-00023-f001:**
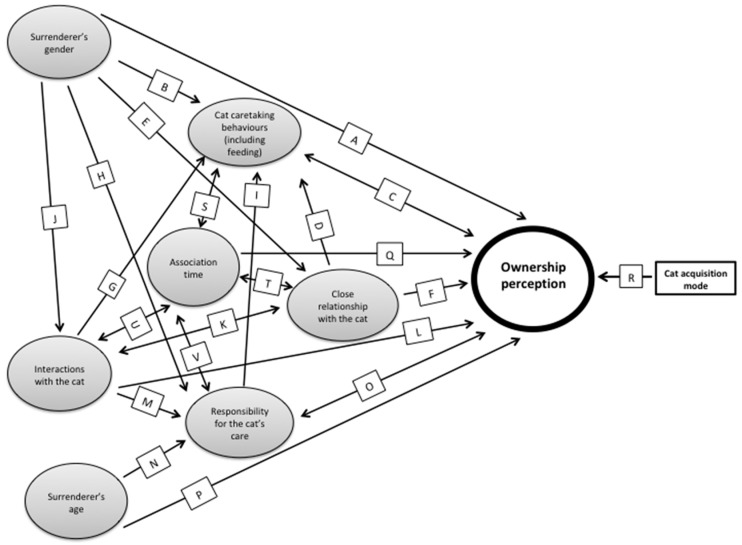
Proposed model of causes of surrenderer’s perceived ownership of passively acquired cats at the time the cat was surrendered; cat acquisition method is included to indicate the strong association between this variable and ownership perception (relationship R).

**Table 1 animals-08-00023-t001:** Questionnaire categories and data variable details.

Categories	Variable Details
*Respondent level data*	
Surrenderer’s demographics	Gender, age, occupation, postcode, socioeconomic indices (Index of relative socio-economic advantage or disadvantage, Index of education and occupation and Index of economic resources [ [25]] of surrenderer’s postcode ^1^).
Surrenderer’s previous cat ownership	Previously owned one or more cats (yes or no).
Surrenderer’s attitudes to cats in general	Degree of agreement with each of “You like cats,” “Cats are good company,” “Cats are independent,” “Cats are low maintenance pets,” and “Cats are expensive pets.”
*Cat level data relating to the surrendered cats*	
Ownership status ^2^	Cats were categorised as “owned” or “unowned” and surrenderers were classified as “owners” or “non-owners” as explained in methods.
Acquisition method	Passive: cat was found, originally a “stray,” a gift, brought home by children, left with them by another person.Active: cat was acquired intentionally from an animal shelter, breeder, pet shop or through a private transaction.
Association time	The time period for which the surrenderer had an association with the surrendered cat.
Responsibility for the cat ^2^	Degree of agreement with the statement “I considered myself to be responsible for the cat’s care.”
Attachment to the cat ^2^	Degree of agreement with the statement “I had a close relationship with the cat.”
Feeding ^3^	The surrenderer’s cat feeding behaviours:Feeding frequency.Feeding duration.
Other caretaking behaviours ^4^	Caretaking behaviours displayed towards the cat by the surrenderer: confining the cat to their property at night (±during the day), buying toys for the cat, having the cat sterilised, vaccinated, or microchipped.
Number of types of caretaking behaviours shown towards the cat (excluding feeding and sterilisation)	A variable was created that summed the number of caretaking behaviours other than feeding and sterilisation that the surrenderer displayed towards the cat out of the four possible behaviours the surrenderer could have displayed. All cats that had missing values for any of the caretaking behaviours were excluded for this variable.
Interactions with the cat ^4^	Interactions that the surrenderer had with the cat: (a) whether the cat was allowed inside the surrenderer’s house; (b) allowed to sleep on the beds and whether the surrenderer (c) held, stroked, cuddled or petted the cat or allowed their children to do so or (d) played with the cat or allowed their children to do so.
Number of types of interactions with the cat	A variable was created that summed the number of types of interactions with the cat that the surrenderer had out of the four possible interaction types. All cats that had missing values for any of the interactions were excluded for this variable.

^1^ These indices are calculated by the Australian Bureau of Statistics for each postcode area in Australia using data from national censuses (conducted every 5 years). We used the indices from the 2011 census, the most recently available at the time of writing. Each index ranks geographic areas across Australia in terms of their relative socio-economic advantage and disadvantage. The Index of relative socio-economic advantage or disadvantage summarises information about the economic and social conditions of people and households within an area. The Index of education and occupation is designed to reflect the educational and occupational level of communities. The Index of economic resources focuses on the financial aspects of relative socio-economic advantage and disadvantage, by summarising variables related to income and wealth. ^2^ These variables were quantified using 5-point Likert scales, from strongly disagree to strongly agree; with the middle category as “neither agree nor disagree.” Items used to describe attachment to the cat were modified from the Lexington Attachment to Pets Scale [26,27]. ^3^ Unowned cats only; it was assumed that all owned cats were fed by their owner. ^4^ For each, surrenderers were asked if they displayed this behaviour towards the cat: and could answer yes, no or unsure.

**Table 2 animals-08-00023-t002:** Distributions of perceived ownership of cats at the time the cat was surrendered by each of cat acquisition method, association time and surrenderer’s relationship with, interactions with and caretaking behaviours shown towards, the surrendered cat (*n* = 96).

**Variable**	**% in Each Category, for Surrenderers Who Considered That They Owned the Cat (Number ^1^)**	**% in Each Category, for Surrenderers Who Considered That They Did Not Own the Cat (Number ^1^)**	***p* Value ^2^**
**Cat acquisition method**			**<0.01**
Active	63% (25/40)	0% (0/56)	
Passive	37% (15/40)	100% (56/56)	
**Duration of surrenderer’s association with surrendered cat (association time)**			**<0.01**
≥3 days to <1 month	8% (3/40)	39% (22/56)	
1 month to <12 months	45% (18/40)	50% (28/56)	
≥12 months	48% (19/40)	11% (6/56)	
**Surrenderer’s degree of agreement with the statements ^3^:**			
*“I considered myself to be responsible for the cat’s care”*			<0.01
Agree	98% (39/40)	70% (39/56)	
Neither agree nor disagree	3% (1/40)	2% (1/56)	
Disagree	0% (0/40)	29% (16/56)	
*“I had a close relationship with the cat”*			<0.01
Agree	75% (30/40)	31% (17/54)	
Neither agree nor disagree	8% (3/40)	7% (4/54)	
Disagree	18% (7/40)	61% (33/54)	
*“I considered the cat to be a friend or companion”*			<0.01
Agree	73% (29/40)	24% (13/54)	
Neither agree nor disagree	8% (3/40)	9% (5/54)	
Disagree	20% (8/40)	67% (36/54)	
*“I considered the cat to be a member of my family”*			<0.01
Agree	75% (30/40)	15% (8/53)	
Neither agree nor disagree	10% (4/40)	15% (8/53)	
Disagree	15% (6/40)	70% (37/53)	
**Surrenderer’s interactions with the surrendered cat**			
*Allowing the cat inside their house*			<0.01
Yes	92% (36/39)	35% (19/55)	
*Allowing the cat to sleep on beds*			<0.01
Yes	76% (29/38)	7% (4/54)	
*Holding, stroking, cuddling or petting the cat or allowing their children to do so*			<0.01
Yes	95% (36/38)	34% (18/53)	
*Playing with the cat or allowing their children to do so*			<0.01
Yes	100% (38/38)	32% (17/53)	
**Caretaking behaviours shown by the surrenderer towards the surrendered cat**			
*Confining the cat to their property at night (±during the day)*			<0.01
Yes	76% (28/37)	14% (7/51)	
*Buying toys for cat*			<0.01
Yes	84% (31/37)	8% (4/52)	
*Having the cat sterilized* ^4^			0.04
Yes	27% (8/30)	6% (2/34)	
*Having the cat vaccinated*			<0.01
Yes	74% (28/38)	2% (1/52)	
*Having the cat microchipped*			<0.01
Yes	59% (22/37)	2% (1/52)	
*Feeding the cat*			<0.01
Yes	100% (40/40) ^5^	57% (31/54)	
**Surrenderer (*n* = 31) who considered that they did not own the cat and who fed the surrendered cat:**			
*Feeding frequency*			
6–7 days every week		90% (28/31)	
5 or less days every week		10% (3/31)	
*Duration of feeding*			
≥3 days to <1 week		6% (2/31)	
1 week to <1 month		35% (11/31)	
≥1 month to <6 months		42% (13/31)	
≥6 months		16% (5/31)	
*Bought food specifically to feed the unowned cat*			
Yes ^6^		55% (17/31)	

^1^ Total number of surrenderers differs between variables because not all surrenderers answered each question; the maximum numbers of responses = 40 for owned cats and 56 for unowned cats. Percentages do not always sum to 100 due to rounding. ^2^
*p*-value for comparison of distributions between surrenderers who considered that they owned the cat and surrenderers who considered that they did not own the cat. ^3^ For each statement, somewhat agree and strongly agree were pooled to form the category “agree” and somewhat disagree and strongly disagree were pooled to form the category “disagree.” Data were not collapsed when calculating *p*-values. ^4^ Includes only cats whose association time was ≥1 month and excludes cats that the surrenderer knew had been sterilised but this was not instituted by the surrenderer; no response was assumed to indicate that the surrenderer did not have the cat sterilized. ^5^ It was assumed that all owned cats were fed. ^6^ No response was assumed to indicate that food was not bought specifically to feed the unowned cat.

**Table 3 animals-08-00023-t003:** Phi coefficients for associations between surrenderers’ cat caretaking behaviours and interactions ^1^.

	Allowing the Cat Inside Their House	Allowing the Cat to Sleep on the Bed	Holding, Stroking, Cuddling or Petting the Cat or Allowing Their Children to do So	Playing with the cat or Allowing Their Children to do So	Confining the Cat to Their Property at Night (±during the Day)	Buying Toys for the Cat	Having the Cat Sterilised	Having the Cat Vaccinated	Having the Cat Microchipped
Allowing the cat to sleep on the bed	0.64								
Holding, stroking, cuddling or petting the cat or allowing their children to do so	0.70	0.64							
Playing with the cat or allowing their children to do so	0.75	0.62	0.93						
Confining the cat to their property at night (±during the day)	0.61	0.71	0.64	0.61					
Buying toys for the cat	0.57	0.81	0.69	0.66	0.71				
Having the cat sterilised	0.26	0.28	0.19	0.27	0.20	0.22			
Having the cat vaccinated	0.55	0.83	0.59	0.57	0.73	0.83	0.35		
Having the cat microchipped	0.52	0.73	0.49	0.48	0.63	0.71	0.36	0.88	
Feeding the cat	0.57	0.45	0.60	0.62	0.48	0.48	0.15	0.42	0.36

^1^ All variables were binary (i.e., yes or no); for all coefficients, Fisher’s exact 2-sided *p*-values were <0.01 except for associations with ‘Having the cat sterilised’ where *p*-values were 0.162 to 0.427 except for Allowing the cat inside their house (0.07), Allowing the cat to sleep on the bed (0.037), Playing with the cat or allowing their children to do so (0.04), Having the cat vaccinated (0.01) and Having the cat microchipped (0.01). Cat caretaking and interaction behaviours were not recorded for some cats; numbers of surrenderers and their cats for coefficients with “Having the cat sterilised” were 58–64 as this variable was used only for cats whose association time was ≥1 month and excluding cats that the surrenderer knew had been sterilised but this was not instituted by the surrenderer; for other coefficients, data were available for between 85 and 92 surrenderers and their cats.

**Table 4 animals-08-00023-t004:** Putative determinants ^1^ of surrenderer’s perceived ownership at time the cat was surrendered for 71 passively acquired cats.

**Variable**	**Number of Cats ^2^**	**% (No.) That Were Identified as Owned**	**Adjusted Odds Ratio ^3^**	**95% Confidence Interval**	***p*-Value ^4^**
**Duration of association with surrendered cat (association time) (*n* = 71)**					**0.20**
≥3 days to <1 month	25	12% (3)	Reference category		
1 month to <12 months	36	22% (8)	2.1	0.5–8.8	0.31
≥12 months	10	40% (4)	5.7	0.9–28.1	0.08
**Surrenderer’s gender (*n* = 70)**					**0.20**
Male	21	14% (3)	Reference category		
Female	49	24% (12)	2.5	0.6–11.2	0.23
**Surrenderer’s age (*n* = 70)**					**0.02**
18–35 years	23	39% (9)	Reference category		
36–55 years	30	13% (4)	0.3	0.1–1.1	0.08
>55 years	17	6% (1)	0.1	0.0–0.8	0.03
**Surrenderer’s degree of agreement with the statements:**					
*I considered myself to be responsible for the cat’s care (n = 71)*					**<0.01**
Strongly disagree	14	0% (0)	Reference category ^5^		
Somewhat disagree	2	0% (0)			
Neither agree nor disagree	2	50% (1)			
Somewhat agree	25	12% (3)	3.0	0.3–33.4	0.37
Strongly agree	28	39% (11)	13.9	1.5–128.3	0.02
*I had a close relationship with the cat (n = 69)*					**<0.01**
Strongly disagree	28	4% (1)	Reference category ^5^		
Somewhat disagree	7	14% (1)			
Neither agree nor disagree	5	20% (1)			
Somewhat agree	17	35% (6)	7.8	1.5–39.5	0.01
Strongly agree	12	50% (6)	12.4	2.1–72.0	0.01
**Surrenderer fed the cat (*n* = 69)**					**<0.01**
No	23	0% (0)	Reference category		
Yes	46	33% (15)	17.0	3.3–∞ ^6^	<0.01
**Number of types of caretaking behaviours shown towards the cat (excluding feeding and sterilisation; maximum = 4) (*n* = 60)**					**<0.01**
0	45	9% (4)	Reference category		
1–2	9	44% (4)	9.8	1.6–60.1	0.01
3 or 4	6	83% (5)	80.3	5.5–1169.7	<0.01
**Number of types of interactions with the cat (maximum = 4) (*n* = 64)**					**<0.01**
0	31	0% (0)	Reference category		
1–2	11	27% 3)	12.1	1.7–∞ ^6^	<0.01
3 or 4	22	41% (9)	26.3	5.0–∞ ^6^	<0.01

^1^ Other than association time and surrenderer’s gender, only those determinants with overall *p*-values of 0.02 or less when adjusted for association time are included in this table. The other eleven potential determinants that were assessed had overall *p*-values when adjusted for association time of 0.34 to 0.96. ^2^ Total number of surrenderers differs between variables because not all surrenderers answered each question and so may not add to 71. ^3^ Univariable analysis for relationship between association time and perceived ownership. All other analyses were adjusted only for association time. ^4^ Bold values are overall *p*-values for variable; non-bolded values are *p*-values for the specific level relative to the reference category. ^5^ Rows with a common vertical line were pooled for analyses. ^6^ ∞ = infinity.

**Table 5 animals-08-00023-t005:** Bases for pathways in a proposed model of causes of a surrenderer’s perceived ownership of passively acquired cats at the time the cat was surrendered (Figure 1).

Pathway ^1^	Basis for Pathway
A	Females are more likely to own a cat than males [ [29]] and to have more intense relationships with cats [30]. Having more intimate knowledge of something leads to a greater sense of ownership [31]. Therefore, if one has an intimate and close relationship with a cat, one is likely to perceive ownership for it.
B	Females provide more caretaking behaviours to cats than males [ [20],[32]].
C	The more time and effort invested in a relationship or object (in this case the time and effort and possibly financial expenditure, invested in cat caretaking behaviours), the greater the feeling of ownership (self-investment) [ [33]]. This relationship is likely to work both ways, as it is plausible that if one perceives ownership for a cat one might be more willing to expend time, financial resources and effort for the cat in the form of caretaking behaviours.
D	Close relationships require investment of time and resources and (in people) the closeness of a relationship has been described by the frequency and strength of the impact each individual has on the other [ [34],[35]], as reported in dog-human relationships [36]. Therefore, if the human-cat relationship is similar, logically, if one has a close relationship one is more likely to invest time and resources into that relationship (caretaking behaviours). However, the type of caretaking behaviours displayed may vary considerably between people, with feeding being the most common behaviour, likely followed by other behaviours that require the expenditure of more effort, time and money.
E	Females are reportedly more likely to be “cat people” [ [37]] than males and are also reported to have more intense relationships with cats [30], suggesting that females are more likely to have a close relationship with a cat.
F	If one has a close relationship with a cat (which implies intimacy) one is likely to develop a sense of ownership. This relationship has a similar basis to that explained in pathway A.
G	Women are more likely to be a cat’s primary caregiver and to be sensitive to the cat’s physical and ethological needs [ [32]]. Females are more likely to be cat semi-owners or colony caretakers than males [12,21]. This suggests that females are more likely to feel a sense of responsibility for a cat’s care than males.
H	Interactions with the cat which contribute to the cat’s sociability and tractability will make it easier to perform caretaking behaviours that involve handling; this may subsequently increase the likelihood of a person performing these behaviours. In addition, the strong relationship reported between the frequency of dog-owner interactions and responsible dog ownership behaviours [ [36]] lends support to the proposed relationship between interactions and caretaking behaviours in cats.
I	Responsibility and caretaking behaviours are frequently associated [ [6],[38]]. Logic suggests that the more responsible one feels for a cat the more likely one is to display caretaking behaviours towards the cat. People who provide caretaking behaviours towards cats in colonies have a sense of responsibility towards the cats [6,38].
J	Cat sociability and time spent with owner are increased with a female owner [ [32]], implying that females interact with cats more than males. Therefore, females may be more likely to show more interaction behaviours towards a cat than males.
K	Close relationships generally require investment of time and resources as explained in pathway D. One can extrapolate that this is similar in human-companion-animal relationships also.Logically the more interactions one has with a cat (if positive) and the more time spent interacting with the cat, especially if having an impact on the cat (i.e., that the cat is dependent on them or displays affection towards them), the closer the relationship with the cat. This relationship is postulated to work both ways.
L	The basis for this pathway follows the same logic as that explained for pathway C.
M	Cat colony caretakers report feeling responsible for the colony cats even though the cats are not their pets and may not be well socialised [ [11]].
N	Responsible behaviour increases with age. Older people display more responsible cat ownership behaviours [ [38]] and it seems logical that this would result from an increased sense of responsibility for the cat’s care.
O	Responsibility and ownership are frequently linked in the literature, with terms such as responsible (cat) ownership and responsible (cat) ownership behaviours frequently used [ [6],[38]].The concept of ownership having responsibilities and of “taking ownership” for something you are responsible for (and vice versa) is common in the literature in a wide variety of fields [39,40,41].
P	Increasing surrenderer age was associated with decreased likelihood of perception of ownership towards surrendered cats.
Q	Increasing length of association with a cat was closely linked to increased likelihood of perception of ownership of that cat [ [7]].
R	In the study sample, active acquisition of the cat predicted ownership perception in 100% of cases. This is consistent with the western perception of ownership in which ownership is associated with the cost and the active acquisition of something.
S	The longer time one is associated with a cat the more time is available to perform caretaking behaviours (especially those that might necessitate some planning or organisation or time to perform such as vaccination). This relationship is postulated to work both ways as it is plausible that the more caretaking behaviours one displays towards a cat the longer one is likely to be associated with the cat while performing the behaviours and because of the time and resources invested in the cat the more likely one might be to continue to associate with the cat (self-investment) [ [33]].
T	It is likely that the more time associated with the cat (reinforced by positive interactions with the cat), the more one is likely to feel that one has a close relationship with the cat. This relationship is postulated to work both ways as the closer your relationship with the cat the more likely you are to spend time with the cat. As described in pathway D, the closeness of inter-human relationships has been described by the frequency of the impact each individual has on the other and it seems likely that this is also the case for inter-species relationships [ [34],[35]].
U	The longer time one is associated with a cat the more time is available to interact with the cat. This pathway is postulated to work both ways as the more interactions one has with a cat the more time one is likely to be associated with the cat. The basis for this pathway is similar to that explained for pathway S.
V	The longer one is associated with a cat the more responsible one is likely to feel for the cat. This relationship is also postulated to also work both ways and have a similar basis to that explained in pathways K and S.

^1^ Shown as an arrow in Figure 1.

**Table 6 animals-08-00023-t006:** Putative determinants ^1^ of whether the surrenderer fed the cat that was surrendered for 53 unowned cats.

**Variable**	**Number of Cats ^2^**	**% (No.) That Were Fed**	**Adjusted Odds Ratio ^3^**	**95% Confidence Interval**	***p*-Value ^4^**
**Surrenderer’s gender (*n* = 52)**					**0.09**
Male	17	35% (6)	Reference category		
Female	35	66% (23)	3.1	0.8–12.5	0.09
**Index of economic resources for surrenderer’s postcode *(n* = 53) ^5^**					**0.13**
1–3	31	58% (18)	Reference category		
4–7	17	65% (11)	1.8	0.4–8.4	0.40
8–10	5	20% (1)	0.2	0.0–1.7	0.09
**Surrendered cat’s age category (*n* = 53)**					**0.07**
Adult (≥6 months)	43	49% (21)	Reference category		
Kitten (<6 months)	10	90% (9)	6.3	0.8–162.9	0.07
**Surrenderer’s degree of agreement with the statement “I am responsible for the cat’s care” (*n* = 53)**					**<0.01**
Strongly disagree	13	0% (0)	Reference category ^6^		
Somewhat disagree	2	100% (2)			
Neither agree nor disagree	1	0% (0)			
Somewhat agree	22	64% (14)	8.9	1.7–70.7	<0.01
Strongly agree	15	93% (14)	36.1	3.7–1129.5	<0.01
**Surrenderer’s degree of agreement with “I had a close relationship with the cat” (*n* = 51)**					**0.05**
Strongly disagree	27	26% (7)	Reference category ^6^		
Somewhat disagree	6	83% (5)			
Neither agree nor disagree	3	100% (3)			
Somewhat agree	11	91% (10)	9.2	1.2–237.0	0.03
Strongly agree	4	75% (3)	2.9	0.2–100.3	0.39
**Number of types of caretaking behaviours shown towards the cat (excluding feeding and sterilisation; maximum = 4) (*n* = 45)**					**0.07**
0	40	45% (18)	Reference category		
1–2	4	100% (4)	6.4	1.0–∞ ^7^	0.07
3 or 4	1	100% (1)			
**Number of types of interactions with the cat (maximum = 4) (*n* = 50)**					**<0.01**
0	30	37% (11)	Reference category		
1–2	8	50% (4)	1.7	0.3–9.4	0.54
3 or 4	12	100% (12)	19.0	3.4–∞	<0.01

^1^ Only those determinants with overall *p*-values of ≤0.13 when adjusted for shelter are included in this table. The other eleven potential determinants of feeding that were assessed had overall *p*-values adjusted for shelter of 0.26 to 0.96. ^2^ Total number of surrenderers differs between variables because not all surrenderers answered each question. All variables have fewer than 53 responses as not all the surrenderers of the 53 cats which were fed answered each question. ^3^ Adjusted only for shelter. ^4^ Bold values are overall *p*-values for variable; non-bolded values are *p*-values for the specific level, relative to the reference category. ^5^ Numbers shown are the decile that the surrenderer’s postcode was in based on its index value; one-tenth of Australian postcodes were in each decile. A lower number indicates that the surrenderer’s postcode area was relatively disadvantaged compared to an area with a higher number [25]. ^6^ Rows with a common vertical line were pooled for analyses. ^7^ ∞ = positive infinity.

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
