# Peer review of "Surrenderers’ Relationships with Cats Admitted to Four Australian Animal Shelters"

_animals, 2018, doi:10.3390/ani8020023_

Round 1

Reviewer 1 Report

This is a very interesting manuscript about a topic that has not yet received a great deal of attention.  The authors clearly describe the importance of better understanding how people who care for cats view themselves.  I have a number of suggestions and requests below.

My main question is the focus of the manuscript.  Is the focus to use the data included here, as well as the literature to develop the causal model or is the causal model just a component of the ownership question?  Based on what is included, I would guess the former, but the abstract, introduction, and sequencing of the manuscript don’t seem to support that.  Please consider and clarify throughout how the different manuscript elements relate to one another and what the primary focus is.

The reference format in the text needs to be tidied up.

Abstract:

Line 22: “were compared” but it isn’t clear what the comparison is.  Please edit.

Line 31: please add a bit more detail to the type of model that is mentioned here.  This is an important part of the work and isn’t really highlighted as much as it should be.

Introduction:

I think that some review of the causal diagram modeling, both the theoretical underpinnings as well as any practical applications that are related to this topic (if they exist) is important to add to the introduction. 

Line 50: I think that this should reference the previous Zito et al publication in the definition of semi-ownership.

Line 51: “are usually left” I think some clarity about this being in Australia or include some other references.

Line 72: “cat caretakers” this has several meanings in the literature. Please clarify how this term is used in this manuscript or pick a different one. Also please clarify how this term is different from semi-owners.  Are all owners Caretakers in this manuscript?  That would be confusing I think.  Are semi-owners the same as caretakers?  Are caretakers anyone who provides any care to a cat and semi-owners people who provide a certain level of care?  Please check terminology here and throughout the manuscript for clarity and consistency.

Paragraph starting line 86: this feels like it could go into the discussion under limitations and then the next paragraph could clarify that this study was a pilot with a convenience sample.

Methods:

Line 119: I had to read this a few times to realize that it was the cat being classified by the owner’s statement…and then the owners being classified by the same statement.  I’m still a bit confused: is any person who is classified as an owner then has an owned cat?  Seems like the later statement is what is actually driving the cat classification.  Please clarify, maybe rearrange this paragraph.

Table 1: socioeconomic indices: how these are found or derived is not well described in the text.  Please explain. There is some duplication between the content of this table and the text in section 2.2.  Perhaps just include in the text any additional definitions or sources of data and leave the descriptions only in the Table.

Line 149: please add to the end of this sentence: based on previous analyses.  That will be clearer.

Line 148: are these similarities also from the previous analysis?  What is the number of cats and kittens included here?  I realize this is closely related to the previous study but this manuscript should be a bit more stand alone.

Section 2.2, first paragraph isn’t really statistical analysis, no statistical testing or summary of data is described here.  Consider relabeling as a sub section.  Was the shelter the respondents came from included in any model?  Was it important visually in the descriptive data?  I do understand the limitations relative to the sample size.

Line 153: please explain your decision to use this correlation instead of something like Kappa. And state the assumptions and later, if those assumptions were met.

Are these associations in Table 2 from the models in 2.2.2?  If not, please add what analysis was used, assumptions and whether they were met to the statistical analysis section.

Paragraph beginning line 173: a bit more detail on why the feedback loops are not included in the references but are being used here and what the implications of that are.  I think that also a brief description of how the directed acyclic graphs are developed would be helpful.  Just a few sentences outline the principals and approaches.

Line 205: This is actually Table 2 I believe.

Table 3: please identify the type of correlation in the title. Please also include some general information about the sample sizes for these correlations.

Footnote 1 table 4: “over” should be “overall”

In Table 5: N and P seem to be opposing statements. Please discuss in the text.

Discussion:

Line 269: “responsible” is a bit judgmental and prone to many different interpretations.  Please rephrase throughout.

Line 325: I think that this sort of diagram is uncommon enough that a specific example of how this could be used to move forward the understanding of ownership and the care of the cats would be really instructive here.  It does also relate to the idea that this diagram is the focus of the manuscript and the rest of the data form part of the support for the development of the diagram.

Author Response

This is a very interesting manuscript about a topic that has not yet received a great deal of attention.  The authors clearly describe the importance of better understanding how people who care for cats view themselves.  I have a number of suggestions and requests below.

My main question is the focus of the manuscript.  Is the focus to use the data included here, as well as the literature to develop the causal model or is the causal model just a component of the ownership question?  Based on what is included, I would guess the former, but the abstract, introduction, and sequencing of the manuscript don’t seem to support that.  Please consider and clarify throughout how the different manuscript elements relate to one another and what the primary focus is.

Authors' responses: Yes, our aim with the model was to use the results of the current study along with other sources of evidence and theoretical logic to develop the causal model. We have added text to the Abstract, and to the end of the Introduction, to ensure this is clear.

The reference format in the text needs to be tidied up.

Authors' responses: References have been revised to meet journal standards

Abstract:

Line 22: “were compared” but it isn’t clear what the comparison is.  Please edit.

Authors' responses: We’ve changed the first part of the sentence to ‘Correlations of caretaking and interactions with surrendered cats were calculated………..’

Line 31: please add a bit more detail to the type of model that is mentioned here.  This is an important part of the work and isn’t really highlighted as much as it should be.

Authors' responses: We have added the following text at this point in the text: "This model consisted of a set of variables proposed as directly or indirectly influencing ownership perception, and arrows betwene these to indicate proposed causal relationships." We have also expanded further on the model methodology in the Introduction.

Introduction:

I think that some review of the causal diagram modeling, both the theoretical underpinnings as well as any practical applications that are related to this topic (if they exist) is important to add to the introduction. 

Authors' responses: In the Introduction, we have added an introduction to the model methodology approach that we used.

Line 50: I think that this should reference the previous Zito et al publication in the definition of semi-ownership.

Authors' responses: We have included this at this piont

Line 51: “are usually left” I think some clarity about this being in Australia or include some other references.

Authors' responses: it now read ‘ but in Australia at least they are usually left’

Line 72: “cat caretakers” this has several meanings in the literature. Please clarify how this term is used in this manuscript or pick a different one. Also please clarify how this term is different from semi-owners.  Are all owners Caretakers in this manuscript?  That would be confusing I think.  Are semi-owners the same as caretakers?  Are caretakers anyone who provides any care to a cat and semi-owners people who provide a certain level of care?  Please check terminology here and throughout the manuscript for clarity and consistency.

Authors' responses: Thank you, this distinction is important to our paper and was inadequately described. We now define here that  a semi-owned cat is one that receives food, medical treatment or shelter from a person who intentionally provides this support but does not consider that they own the cat [6] (l 60-62). We have now included a definition (L 57-58) for caretakers as those providing support for a cat (which may include owners and semi-owners).

Paragraph starting line 86: this feels like it could go into the discussion under limitations and then the next paragraph could clarify that this study was a pilot with a convenience sample.

Authors' responses: this topic is covered in more detail in the penultimate paragraph in the Discussion beginning (line 2247). Therefore we have left it here to introduce to the reader why we used a convenience sampling technique. We have added a section heading in the Discussion to distinguish these latter parts which discuss limitations to the study. We have also added in the final paragraph of the Method that this study was a pilot with a convenience sample, as suggested.

Methods:

Line 119: I had to read this a few times to realize that it was the cat being classified by the owner’s statement…and then the owners being classified by the same statement.  I’m still a bit confused: is any person who is classified as an owner then has an owned cat?  Seems like the later statement is what is actually driving the cat classification.  Please clarify, maybe rearrange this paragraph.

Authors' responses: We have edited and added text to clarify this paragraph, as follows:

Each cat was classified as either owned or unowned by the surrender based on the surrender’s level of agreement on a five point Likert scale with the statement “I consider myself to be the owner of the cat”. As the distribution of these responses was bimodal and highly polarised with few responses in the middle of the scale, the Likert scale responses were converted to a dichotomous variable for analyses; cats with a surrenderer who strongly or somewhat agreed with the statement were considered owned by the surrender, and cats with a surrenderer who did not agree (respondents who neither agreed nor disagreed or strongly or somewhat disagreed with the ownership statement) were considered to be not owned by the surrender. For cats classified as owned by the surrender and cats classified as not owned by the surrender, the surrenderer was respectively refered to as the ‘owner’ or the 'non-owner' .

Table 1: socioeconomic indices: how these are found or derived is not well described in the text.  Please explain.

Authors' responses: We have added a footnote to Table 1 to explain the socioeconomic indices.

There is some duplication between the content of this table and the text in section 2.2.  Perhaps just include in the text any additional definitions or sources of data and leave the descriptions only in the Table.

Authors' responses: the only duplication is repetition of the variables that another reviewer requested.

Line 149: please add to the end of this sentence: based on previous analyses.  That will be clearer.

Authors' responses: We’ve added this to this sentence

Line 148: are these similarities also from the previous analysis?  What is the number of cats and kittens included here?  I realize this is closely related to the previous study but this manuscript should be a bit more stand alone.

Authors' responses: Yes, this is also from the previous analysis, hence the statement ‘based on previous analyses’ is valid.

Section 2.2, first paragraph isn’t really statistical analysis, no statistical testing or summary of data is described here.  Consider relabelling as a sub section. 

Authors' responses: Good point. We have edited to address this, including moving some material back to the previous section and creating a new sub-heading ("2.2.1. Pooling of data").

Was the shelter the respondents came from included in any model?  Was it important visually in the descriptive data?  I do understand the limitations relative to the sample size.

Authors' responses: Another good point. Prompted by your comment (and the related comment by reviewer 4), we have done quite a lot more analyses/reanalyses including looking at effects of shelter. Descriptions of these are included in extensive edits of Methods, and Results.

Line 153: please explain your decision to use this correlation instead of something like Kappa. And state the assumptions and later, if those assumptions were met.

Authors' responses: Kappa assesses the extent of agreement above that expected by chance. Our aim was to assess correlation rather than agreement. Agreement would have described the extent to which the same results occurred for each of the pair of variables. As the pairs of variables measured different constructs, it is not useful to assess whether the results agree to a greater extent than that expected by chance. Rather, our interest was simply whether 'Yes' for variable 2 was more (or less) likely for those cats whose respondents who said 'Yes' for variable 1, relative to cats whose respondents who said 'No' for variable 1.

We originally used tetrachoric correlation coefficients for this. But prompted by your query, we have reconsidered this and have redone these analyses using the phi coefficient. The tetrachoric correlation coefficient estimates what the correlation between variables would be if the variables had been measured on a continuous scale. As there is no reason to think of these variables as having an underlying (i.e. latent) continuous distribution, the phi coefficient is more appropriate. This is a measure of correlation for two binary variables. It is interpreted in a similar way to Pearson's correlation coefficient for continuous data. (In fact, for pairs of binary variables as in the current study, the formula used for calculating Pearson's correlation coefficient calculates the phi coefficient.)

Are these associations in Table 2 from the models in 2.2.2?  If not, please add what analysis was used, assumptions and whether they were met to the statistical analysis section.

Authors' responses: We had neglected to describe the stats methods for the results in Table 2 - thank you for picking this up. These methods are now described under "Statistical analyses".

Paragraph beginning line 173: a bit more detail on why the feedback loops are not included in the references but are being used here and what the implications of that are. 

Authors' responses: We have added the following explanation:

"We depicted feedback loops (eg where, for example, variable A affects variable B, which in turn, affects variable A) as two-directional arrows. This was done solely to simplify the appearence of the model. In directed acyclic graphs, feedback loops would instead be incoporpated by showing variable A at timepoint 1 affects variable B at timepoint 2, which in turn, affects variable A at timepoint 3 and so on."

I think that also a brief description of how the directed acyclic graphs are developed would be helpful.  Just a few sentences outline the principals and approaches.

Authors' responses: Good suggestion. We have expanded the description of the model development process considerably.

Line 205: This is actually Table 2 I believe.

Authors' responses: We’ve changed it to Table 2

Table 3: please identify the type of correlation in the title. Please also include some general information about the sample sizes for these correlations.

Authors' responses: The type of correlation has been added to the title. Numbers of cats are shown in the table footnotes.

Footnote 1 table 4: “over” should be “overall”

Authors' responses: Corrected

In Table 5: N and P seem to be opposing statements. Please discuss in the text.

Authors' responses: They are not really opposing, N refers to cat ownership behaviours and P the perception of cat ownership. Older people may show more behaviours but not recognise themselves as owners. 

Discussion:

Line 269: “responsible” is a bit judgmental and prone to many different interpretations.  Please rephrase throughout.

Authors' responses: As reported in Table 5 ‘O’ responsible is a word regularly used in the literature in connection with ownership. Some of the use of this word in this manuscript cannot be altered as it was the word we used in our survey, but we have changed it at the start of the Discussion to purposeful. We use it also to refer to people being responsible for cat care, which is different, and we presume more acceptable, than here where we are referring to responsible behaviour, which is admittedly judgemental.

Line 325: I think that this sort of diagram is uncommon enough that a specific example of how this could be used to move forward the understanding of ownership and the care of the cats would be really instructive here.  It does also relate to the idea that this diagram is the focus of the manuscript and the rest of the data form part of the support for the development of the diagram.

Authors' responses: Good suggestion. We have added that here.

Reviewer 2 Report

In the present paper, the Authors aimed to describe surrenderers’ caretaking behaviours towards, and interactions 
with, the cats they surrender, and to assess associations between potential determinants of perceived ownership,
and the surrenderer’s perceived ownership of the cat at the time of surrender. 

The work appears adequately conducted in accordance with the ethical principles and the results reported are pertinent.

The objectives of the trial are of interest and fit well within the scope of the journal and, in overall, the experiment was performed with an adequate description of methodology used.

In my opinion, the manuscript could be accepted for publication in Animals.

Author Response

In the present paper, the Authors aimed to describe surrenderers’ caretaking behaviours towards, and interactions 
with, the cats they surrender, and to assess associations between potential determinants of perceived ownership,
and the surrenderer’s perceived ownership of the cat at the time of surrender. 

The work appears adequately conducted in accordance with the ethical principles and the results reported are pertinent.

The objectives of the trial are of interest and fit well within the scope of the journal and, in overall, the experiment was performed with an adequate description of methodology used.

In my opinion, the manuscript could be accepted for publication in Animals.

Authors' responses: Thank you.

Reviewer 3 Report

Summary

The aim of the paper was to utilize a previously collected and published data set to assess the attitudes of surrenderers of cats to animal shelters in Australia and determine if there were factors that were associated with ownership. The study then went on to propose a model of such interactions to better define the relationships. Overall, the study found that length of association and active acquisition of the cat were significantly associated with ownership perception. Additionally, this study demonstrated that a large number of “stray” cat surrenders come from semi-ownership situations. This knowledge provides sheltering organizations with the ability to target certain communities and forms of non-ownership to reduce shelter intake.

Broad comments:

Introduction

-          Would be beneficial to briefly list a few of the ethical concerns these cats cause.

Materials and methods:

-          Why was no statistical comparison performed to assess for differences in the four shelter locations?

-          Should be some discussion about the exposures and how the reference categories were selected. For example for the variables duration of association and surrenderers’ age. It would also be useful to know why these three age categories were selected for analysis. This may have occurred in the previous paper but it is very applicable to your results in this one as well.

Results:

-          Why was a p<0.3 determined to be the cut-off for reporting variables and inclusion in modeling?

-          For the variables in Table 4, are these the models that also included duration of association? If so that needs to be clarified in the results as well i.e. “Associations between potential determinants and ownership perception of the cat at the time of surrender, adjusted for duration of association, with overall p-values….” In the Materials and methods you discuss accounting for this variable in modeling but it is unclear in the results which models accounted for this.

-          For pathway N, your argument is that responsible behavior increases with age, but the laws that you cite more-so demonstrate that responsibilities increase with age. Is there evidence that supports that older people act more socially responsible than younger people?

Discussion

-Again you introduce the idea that semi-ownership affects cat welfare but do not give specific examples of how cat welfare is impaired.

-You discuss the addition of caretaking behavior questions into entry interviews for cat surrenders. How will the length of these forms effect collection of data, accuracy of the data, shelter intake practices? Is there any information available on how accurate these forms and data collected from them is? There are several papers looking at the reliability of information collected from surrenderers, consider looking at the data collected on intake for behavioral issues and it’s reproducibility as support for this collection method.

-In discussion of your results being temporal, do you think that a surrenderers’ perception of ownership could also be tied to being perceived by the shelter as the abandoning owner vs the good Samaritan turning in a stray? Do your shelters charge relinquishment fees that could also contribute to an individual’s self-identification? Could these things have effected your surveying?

-Do you feel that building your model of ownership was effected by or biased by the fact that all cats were surrendered and thus you do not include the population of owners, semi-owners, and non-owners who would not surrender a cat? Might these two different population have completely different models of ownership establishment?

-There is no discussion of how your findings differ from previous findings. For example you found that being female was significantly associated with feeding cats, but gender was not significantly associated with perceived ownership. From your modeling it appears that this differs from previous research. Can you postulate why?

Specific comments:

Introduction:

-          Line 58 – “In the United States, currently 27% of…”

-          Line 60 – “direct support from humans. It is….”

-          Line 68 – “and cat curfews will only have an impact on…”

-          Citations are not uniform throughout the paper. The introduction and methods 2.1 study design cites both a number and then authors names in () with the exception of line 62 where the numbers are missing, this then changes in section 2.2, but then is inconsistent again on line 175-176. Per the author guidelines, “In the text, reference numbers should be placed in square brackets [ ], and placed before the punctuation; for example [1], [1–3] or [1,3]. For embedded citations in the text with pagination, use both parentheses and brackets to indicate the reference number and page numbers; for example [5] (p. 10). or [6] (pp. 101–105).

Materials and methods:

-          Lines 158-160, and the sentence in the middle of 161 are results and belong in that section not materials and methods

-          Lines 173 – 180 should be a separate section 2.3 Modelling and come after all of the statistical analyses section

Results:

-          Table 5 pathway F “This relationship has a similar basis to that explained…..”

Author Response

Summary

The aim of the paper was to utilize a previously collected and published data set to assess the attitudes of surrenderers of cats to animal shelters in Australia and determine if there were factors that were associated with ownership. The study then went on to propose a model of such interactions to better define the relationships. Overall, the study found that length of association and active acquisition of the cat were significantly associated with ownership perception. Additionally, this study demonstrated that a large number of “stray” cat surrenders come from semi-ownership situations. This knowledge provides sheltering organizations with the ability to target certain communities and forms of non-ownership to reduce shelter intake.

Broad comments:

Introduction

-          Would be beneficial to briefly list a few of the ethical concerns these cats cause.

Authors' responses: we have added the following: ethical concerns of the euthanasia of many healthy animals, moral stress for the people involved, financial costs to organisations that manage unwanted cats, environmental costs, wildlife predation, potential for disease spread, community nuisance, and welfare concerns for the cats

Materials and methods:

-          Why was no statistical comparison performed to assess for differences in the four shelter locations?

Authors' responses: Good point. Prompted by your comment (and the related comment by reviewer 1), we have done quite a lot more analyses/reanalyses including looking at effects of shelter. Descriptions of these are included in extensive edits of Methods, and Results.

-          Should be some discussion about the exposures and how the reference categories were selected. For example for the variables duration of association and surrenderers’ age. It would also be useful to know why these three age categories were selected for analysis. This may have occurred in the previous paper but it is very applicable to your results in this one as well.

Authors' responses: We categorised each continuous exposure variable into three categories before performing analyses, choosing category cutpoints that both ensured atleast modest numbers of cats in each category and were 'neat' (eg for association time, 1 month and 12 months were used as cutpoints while for surrender's age, 35 and 55 were used). This text has been added to Methods.

Results:

-          Why was a p<0.3 determined to be the cut-off for reporting variables and inclusion in modeling?

Authors' responses: With our reanalyses, we no long used a cutpoint of P<0.3. In the Results text, we now fully explain the basis for including results in Tables 4 and 6. This is also described using footnotes in these Tables (footnote 1 in both cases).

-          For the variables in Table 4, are these the models that also included duration of association? If so that needs to be clarified in the results as well i.e. “Associations between potential determinants and ownership perception of the cat at the time of surrender, adjusted for duration of association, with overall p-values….” In the Materials and methods you discuss accounting for this variable in modeling but it is unclear in the results which models accounted for this.

Authors' responses: Good point. Now added to the Results text. Ditto for results now adjusted for shelter.

-          For pathway N, your argument is that responsible behavior increases with age, but the laws that you cite more-so demonstrate that responsibilities increase with age. Is there evidence that supports that older people act more socially responsible than younger people?

Authors' responses: We have removed our evidence at this point, thank you for your useful comment. There is plenty of evidence of increasing moral behaviour with age, which is the basis for Kohlberg’s theory. Although criticised for some of the detail (e.g. tested more on men than women), the basic principle that moral behaviour increases over time is well supported.

Discussion

-Again you introduce the idea that semi-ownership affects cat welfare but do not give specific examples of how cat welfare is impaired.

Authors' responses: we have added welfare examples: ‘potential welfare issues for cats, such as supplying food that is variable in quantity and quality, attracting cats to high traffic areas, and encouraging the production of unwanted kittens, which together with the cat may have to be surrendered to shelters [6,7]’

-You discuss the addition of caretaking behavior questions into entry interviews for cat surrenders. How will the length of these forms effect collection of data, accuracy of the data, shelter intake practices? Is there any information available on how accurate these forms and data collected from them is? There are several papers looking at the reliability of information collected from surrenderers, consider looking at the data collected on intake for behavioral issues and it’s reproducibility as support for this collection method.

Authors' responses: We could only find one reports of the accuracy of data collected at intake and suggest that this will be highly variable between shelters and possibly between different times of year. We have searched on Google and Web of Science for such evidence. We have added the following to the discussion at the point where interviews are mentioned.

However, it must be acknowledged that lengthening the form may reduce the accuracy of the data, which has been questioned for companion animals being surrendered to shelters on the grounds of the owner’s desire to provide a socially acceptable reason for relinquishment, or to improve the chances of admittance to the shelter and subsequent adoption. Fees may be charged for relinquishment of owned cats but not strays, which may also affect data accuracy.

-In discussion of your results being temporal, do you think that a surrenderers’ perception of ownership could also be tied to being perceived by the shelter as the abandoning owner vs the good Samaritan turning in a stray? Do your shelters charge relinquishment fees that could also contribute to an individual’s self-identification? Could these things have effected your surveying?

Authors' responses: We have addressed this in the previous addition.

-Do you feel that building your model of ownership was effected by or biased by the fact that all cats were surrendered and thus you do not include the population of owners, semi-owners, and non-owners who would not surrender a cat? Might these two different population have completely different models of ownership establishment?

Authors' responses: Yes, this is most certainly a possibility and your comment has made us realise that we'd failed to make this point. We now make this point in the Discussion.

-There is no discussion of how your findings differ from previous findings. For example you found that being female was significantly associated with feeding cats, but gender was not significantly associated with perceived ownership. From your modeling it appears that this differs from previous research. Can you postulate why?

Authors' responses: You are right, the literature supports a strong gender bias in cat ownership and supporting behaviour, as in Table 5:

Females are more likely to own a cat than males [35] and to have more intense relationships with cats [36]. Having more intimate knowledge of something leads to a greater sense of ownership [37], therefore, if one has an intimate and close relationship with a cat one is likely to perceive ownership for it.

Females provide more caretaking behaviours to cats than males [15,38].

Females are reportedly more likely to be “cat people” [43] than males and are also reported to have more intense relationships with cats [36], suggesting that females are more likely to have a close relationship with a cat.

Women are more likely to be a cat’s primary caregiver and to be sensitive to the cat’s physical and ethological needs [38]. Females are more likely to be cat semi-owners or colony caretakers than males [11,16]. This suggests that females are more likely to feel a sense of responsibility for a cat’s care than males.

We have added discussion on this issue:

‘However, contrary to expectations from the extant literature on cat ownership (Table 5), women were not more likely to claim ownership of passively acquired cats than men (Table 4). This may be due to their sense of deference to the attitudes of men on this issue, a factor previously acknowledged in their attitudes to animal welfare (Phillips et a;l., 2010). In this respect, attitudes of men may be more negative towards their female partner owning passively-acquired cats than actively-acquired cats.’

Phillips, C. J. C., Izmirli, S., Kennedy, M., Lee, G.H., Lund, V., Mejdell, C., Pelagic, V.R., Rehn, T., Aldavood, J., Alonso, M., Choe, B.I., Hanlon, A.J., Handziska, A., Illmann, G. and Keeling, L. 2010. An international comparison of female and male students’ attitudes to the use of animals. Animals 1, 7-26.

Specific comments:

Introduction:

-          Line 58 – “In the United States, currently 27% of…”

-          Line 60 – “direct support from humans. It is….”

-          Line 68 – “and cat curfews will only have an impact on…”

Authors’ response: we have made all of these changes

-          Citations are not uniform throughout the paper. The introduction and methods 2.1 study design cites both a number and then authors names in () with the exception of line 62 where the numbers are missing, this then changes in section 2.2, but then is inconsistent again on line 175-176. Per the author guidelines, “In the text, reference numbers should be placed in square brackets [ ], and placed before the punctuation; for example [1], [1–3] or [1,3]. For embedded citations in the text with pagination, use both parentheses and brackets to indicate the reference number and page numbers; for example [5] (p. 10). or [6] (pp. 101–105).”

Authors' responses: Citations have been changed to Animals format.

Materials and methods:

-          Lines 158-160, and the sentence in the middle of 161 are results and belong in that section not materials and methods

Authors' responses: This text was to explain why associations between each potential determinant of ownership and perceived ownership of cats at the time the cat was surrendered were assessed using only respondents surrendering passively acquired cats. However we have abbreviated this and moved the details to Results.

-          Lines 173 – 180 should be a separate section 2.3 Modelling and come after all of the statistical analyses section

Authors' responses: Sub-section added ("2.2.4. Model development").

      Results:

-          Table 5 pathway F “This relationship has a similar basis to that explained…..”

Authors' responses: changed

Reviewer 4 Report

Thank you for the opportunity to read this manuscript.  That people relinquish cats, and may or may not have a relationship with the cat, was what I took to be the rationale for the paper.  The paper was fairly dense and lengthy, with a number of tables and analyses.  The bottom line conclusions did not really seem to add a great deal to knowledge or provide realistic directions for future research or actions.  I can see that a great deal of effort has gone into this paper, so perhaps there might be a way of simplifying the analyses and making this a different manuscript.

Author Response

Thank you for the opportunity to read this manuscript.  That people relinquish cats, and may or may not have a relationship with the cat, was what I took to be the rationale for the paper.  The paper was fairly dense and lengthy, with a number of tables and analyses.  The bottom line conclusions did not really seem to add a great deal to knowledge or provide realistic directions for future research or actions.  I can see that a great deal of effort has gone into this paper, so perhaps there might be a way of simplifying the analyses and making this a different manuscript.

Authors' responses: We report for the first time the characteristics of semi-ownership and the major reasons why owners identify as such. This is useful for shelters to identify owners and target intervention to reduce the numbers of unwanted cats.

We agree that the modelling is complex and hopefully revisions that we have introduced will make it easier to understand. The Simple Summary is available for anyone who wants none of the detail of the modelling work undertaken, but for those more scientifically orientated, we agree with three other reviewers that the modelling details must be available.